

# Quantum Yields of CHDO above 300 nm

Ernst-Peter Röth[1], Luc Vereecken[2]

[1]Institute for Energy and Climate Research IEK-7: Stratosphere, Forschungszentrum Jülich GmbH, 52425 Jülich, Germany

[2]Institute for Energy and Climate Research IEK-8: Troposphere, Forschungszentrum Jülich GmbH, 52425 Jülich, Germany

Corresponding author : Ernst-Peter Röth     e.p.roeth@fz-juelich.de



**Abstract :** *The photolysis of mono-deuterated formaldehyde, CHDO, is a critical process in the deuterium-enrichment of stratospheric hydrogen formed from methane. A consistent description of the quantum yields of the molecular and radical channels of the CHDO photolysis is deduced from literature data. The fluorescence measurements of Miller and Lee (1978) provided a first data set to deduce the product quantum yields. An alternative analysis is provided by the measured quantum yield spectrum for the radical channel of the $CD_2O$ photolysis by McQuigg and Calvert (1969), which is corrected for wavelength dependency and combined with the $CH_2O$ quantum yield spectrum to provide an approximation for CHDO. Both approaches provide consistent results. Finally, the findings of Troe (1984, 2007) enable the specification of the pressure dependence of the quantum yield for $CH_2O$ and $CD_2O$ and, hence, for CHDO. We find that the radical channel does not show a pressure dependence, whereas the molecular channel is dominated by tunneling and quenching processes. For modeling purposes, simplified representations are given, and as an example for their application, the altitude dependence of the ratio of $J(CHDO \rightarrow HD+CO)$ and $J(CH_2O \rightarrow H_2+CO)$ is provided.*



**1. Introduction**
Measurements over the last decades showed that molecular hydrogen, $H_2$, in the stratosphere
is enriched in deuterium compared to $H_2$ in the troposphere (see e.g.: Ehhalt and Volz, 1976;
Gerst and Quay, 2001; Rahn et al., 2003; Rice et al., 2003; Röckmann et al., 2003; McCarthy
et al., 2004; Rhee et al., 2006). Gerst and Quay (2001) suggested that this enrichment could be
due to the differential isotope fractionation in the photo-oxidation of methane. Measurements
of the vertical profiles of the isotope content in $H_2$ and $CH_4$, available since 2003, allowed the
interpretation and modeling of the observed enrichment (see e.g. Pieterse et al., 2011). The
methane photo-oxidation consists of various reaction steps, each of which contribute kinetic
isotope effects, KIE, that have to be considered (e.g. Feilberg et al., 2005; Mar et al., 2007).
The last but critical step in the reaction chain to produce the hydrogen isotope D from the
mono-deuterated isotopologue of formaldehyde, CHDO is its photolysis.
Compared to $CH_2O$, the available data for the mono-deuterated isotopologue CHDO are
scarce. Only its spectrum was measured (c.f. Mainz Spectral Atlas, Keller-Rudek and
Moortgat, 2021). The quantum yields for the molecular and the radical fragmentation
branches of the CHDO photolysis, as well as the rate constants for the quenching reactions
were not measured at all or with insufficient accuracy. Thus, despite its importance for the
atmospheric production of HD, the photolysis of CHDO is still poorly defined; at this time, it
is the most uncertain factor in the overall fractionation of formaldehyde. For example, the
measured or estimated fractionation factors for the molecular channel range from 1.08 to 1.82
(e.g. Feilberg et al., 2005; Rhee et al., 2006, Mar et al., 2007; Nilsson et al., 2009; Röckmann
et al., 2010). Moreover, the measurements by Nilsson et al. (2009) are the only ones
considering the pressure dependence of the fractionation factor due to reactions R3, R4, and
R7 (see Table 1).
In this work, we aim to provide information for the modeling of CHDO photochemistry for
atmospheric conditions, i.e. for a limited domain of temperature and pressure, by deducing
$\Phi^{mol}$ and $\Phi^{rad}$ for CHDO from literature information, based on the scant data available and
supplemented by a number of plausible assumptions. We do this based on two approaches: the
first is based on the fluorescence measurements of Miller and Lee (1978) and literature data
on energy transitions (e.g. Yeung and Moore, 1973; Chuang et al., 1987; Osborn, 2008; Fu et
al. 2011). The second approach assumes that the measurements of McQuigg and Calvert



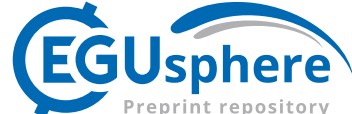

35 (1969) can be corrected via the comparison of the $CH_2O$ measurement with later experiments

36 (see e.g. the overview by Röth and Ehhalt, 2015).



39 **2. Photolysis reaction mechanism**


41 Based on the available literature (e.g.: Aràujo et al., 2009; Breuer and Lee, 1971; Chuang et

42 al., 1987; Yamaguchi et al., 1998) we propose a photolytic reaction scheme of CHDO in

43 Table 1, analogous to that of $CH_2O$ (Röth and Ehhalt, 2015). The scheme involves a

44 cascading series of fragmentation channels competing with stepwise quenching by collisional

45 energy loss, starting at the excited singlet state $S_1$. Reactions via the triplet state of CHDO are

46 not considered here, as they are only accessible at wavelengths below 300 nm (Aràujo et

47 al.,2009), while we concentrate on wavelengths above this limit in this work.

48

49 **Table 1 :** Reaction scheme of the photolysis of CHDO. The asterix * stands for excitations

50 able to lead to bond breaking, whereas the index # indicates lower energies and lead

51 ultimately to thermalized CHDO.

52 $CHDO(S_0) + h\upsilon \rightarrow CHDO^*(S_1)$          (R0)

53    $CHDO^*(S_1) \rightarrow CHDO^\# + h\nu_1$      (R1)

54    $CHDO^*(S_1) \rightarrow CHDO^*(S_0)$      (R2)

55      $CHDO^*(S_0) \rightarrow H+CDO/D+CHO$   (R2a)

56      $CHDO^*(S_0) \rightarrow CO + HD$    (R2b)

57      $CHDO^*(S_0) + M \rightarrow CHDO^\#(S_0)$  (R2c)

58    $CHDO^*(S_1) + M \rightarrow CHDO^{*-\Delta\varepsilon1}(S_0)$   (R3)

59      $CHDO^{*-\Delta\varepsilon1}(S_0) \rightarrow H+CDO/D+CHO$  (R3a)

60      $CHDO^{*-\Delta\varepsilon1}(S_0) \rightarrow CO + HD$   (R3b)

61      $CHDO^{*-\Delta\varepsilon1}(S_0) + M \rightarrow CHDO^\#(S_0)$  (R3c)

62    $CHDO^*(S_1) + M \rightarrow CHDO^{*-\Delta\varepsilon2}(S_1)$   (R4)

63      $CHDO^{*-\Delta\varepsilon2}(S_1) \rightarrow CHDO^\#(S_0) + h\nu_5$  (R5)

64      $CHDO^{*-\Delta\varepsilon2}(S_1) \rightarrow CHDO^{*-\Delta\varepsilon2}(S_0)$  (R6)

65        $CHDO^{*-\Delta\varepsilon2}(S_0) \rightarrow H+CDO/D+CHO$  (R6a)

66        $CHDO^{*-\Delta\varepsilon2}(S_0) \rightarrow CO + HD$  (R6b)

67        $CHDO^{*-\Delta\varepsilon2}(S_0) +M \rightarrow CHDO^\#(S_0)$  (R6c)

68    $CHDO^{*-\Delta\varepsilon2}(S_1) +M \rightarrow$  …     (R7)




After excitation of the ground state CHDO($S_0$) (R0) by a photon of a given wavelength, the
excited reaction product CHDO$^*$($S_1$) decays by fluorescence (R1), or transitions to the $S_0$
ground state surface as an excited CHDO* molecule with either all available energy (R2) or
with a variable amount of energy -$\Delta\varepsilon 1$ lost by quenching (R3). These excited CHDO*($S_0$) and
CHDO*$^{-\Delta\varepsilon 1}$($S_0$) can in turn be quenched by the bath gas in a cascading series (R2c, R3c, R6c),
at each energy level competing with fragmentation to radicals H+CDO/D+CHO (R2a,
R3a,R6a) or to molecular products CO+HD (R2b, R3b,R6b), as described for $CH_2O$ by
Yeung and Moore, (1973). Alternatively, the excited CHDO*($S_1$) can lose an amount of
energy by quenching, but remain on the $S_1$ excited electronic surface (R4). This state can then
undergo processes as above, i.e. decay by fluorescence (R5), transition to the $S_0$ ground state
without (R6) or with (R7) energy loss by quenching, where once again it can undergo further
quenching (R6c) in competition with fragmentation (R6a, R6b). Overall, this scheme
represents a cascading series of quenching steps competing against decomposition and
fluorescence. Only the first few steps in the cascade are represented, but more cascading steps
are possible at lower internal energies. According to the analysis of the fluorescence
measurements by Miller and Lee (1978), these lower-energy reactions are not critical and
need not be considered in detail. Here, R7 simply represents the summation of all subsequent
cascades, from which negligible channels such as *e.g.* the fluorescence channels are omitted.

The quantum yield $\Phi^{rad}$ represents the combined fragmentation to radicals (R2a, R3a, R6a),
while summed fragmentation through the molecular branches (R2b, R3b, R6b) is described by
the quantum yield $\Phi^{mol}$. The total photolysis quantum yield $\Phi^{tot}$, i.e. the decay of excited
formaldehyde into products other than its ground-state, can be derived from the observed CO
production, where CDO and CHO radical fragments react with $O_2$ to form CO and $HO_2$ /
$DO_2$. The quantum yield of the fluorescence is always less than 1% (Miller and Lee, 1978)
and is omitted henceforth.

$$\Phi^{tot} = \Phi^{mol} + \Phi^{rad} \qquad \text{(F1)}$$

Obviously, the sum of $\Phi^{tot}$ and $\Phi^{quench}$, the summed yield of the quenching reactions (R2c,
R3c, R6c), must equal 1 at any wavelength h$\nu$.

$$\Phi^{tot} + \Phi^{quench} = 1 \qquad \text{(F2)}$$






**3. Analysis of fluorescence measurements**

From the fluorescence measurements of Miller and Lee (1978) the quantum yields of both the
fluorescence and the total non-CHDO products can be derived: as shown in Figure 4a, the
contribution of the second step in the reaction cascade is small at low pressure, so we assume
that Table 10 provided by these authors directly gives the reaction rate constants $k_1$ and $k_2$,
where $k_1$ equals the reciprocal lifetime $\tau_{radiation}$ listed and $1/k_2$ is the non-radiative lifetime.
Similarly, the constants $k_5$ and $k_6$ are determined by the lifetimes of the next lower vibrational
level.

The reaction constants $k_3$, $k_4$, and $k_7$ can be deduced from the pressure dependence of the
CHDO fluorescence quantum yield in the Table 2 of Miller and Lee (1978). In the present
paper only the quantum yields at pressures above 1 Torr are considered, where the Ar bathgas
used is assumed to have similar collisional properties as air (Hirschfelder et al.,1954). For
each wavelength the pressure dependence of the data is fitted by a Simplex algorithm
according to Nelder and Mead (1965) by formula F3 for the fluorescence quantum yield $\Phi_F$.
$$\Phi_F = \frac{k_1}{\alpha} + \frac{k_4[M]}{\alpha} \cdot \frac{k_5}{\beta}$$    (F3)
with   $\alpha = k_1 + k_2 + k_3[M] + k_4[M]$    and $\beta = k_5 + k_6 + k_7[M]$

The corresponding reaction constants are listed here in Table 2. With this data set the
experimental fluorescence measurements are well fitted as shown in Figure 1 where, to
improve the clarity of the fit, only the pressure dependent part $\theta(M)$ of equation F3 is plotted
vs pressure:
$$\theta(M) = \frac{k_1}{\phi_F} - )$$    (F4)













**Table 2 :** Results of the least square fit of the quantum yields of CHDO (Miller and Lee
(1978). $k_1$, $k_2$ and $k_5$, $k_6$ are literature data (Miller and Lee, 1978), $k_3$, $k_4$, and $k_7$ are deduced
from these data.

| Wavelength [nm] | $k_1$ [$10^5 s^{-1}$] | $k_2$ [$10^8 s^{-1}$] | $k_3$ [$10^{-11} cm^3\ s^{-1}$] | $k_4$ [$10^{-11} cm^3\ s^{-1}$] | $k_5$ [$10^5 s^{-1}$] | $k_6$ [$10^8 s^{-1}$] | $k_7$ [$10^{-12} cm^3\ s^{-1}$] |
|---|---|---|---|---|---|---|---|
| 314 | 3.03 | 1.79 | 29.7 | 4.59 | 2.78 | 0.50 | 0.57 |
| 318 | 2.50 | 1.32 | 15.4 | 3.48 | 2.50 | 0.40 | 1.15 |
| 326 | 2.78 | 0.50 | 10.9 | 1.77 | 3.57 | 0.22 | 1.79 |
| 329 | 6.41 | 0.50 | 10.4 | 3.95 | 3.00 | 0.20 | 2.06 |
| 330 | 2.50 | 0.40 | 4.81 | 1.05 | 2.44 | 0.13 | 1.35 |
| 338 | 3.57 | 0.22 | 4.89 | 0.84 | 3.45 | 0.07 | 0.77 |
| 344 | 2.44 | 0.13 | 5.95 | 2.78 | 2.40 | 0.06 | 1.39 |
| 353 | 3.45 | 0.07 | 2.38 | 0.76 | 4.00 | 0.03 | 1.24 |


The energy transferred in reaction R2 is either quenched to form a stable molecule
CHDO$^{\#}$($S_0$) or used to drive fragmentation to molecular (CO + HD) or radical products
(H+CDO / D+CHO). Hence, the reactions R2a and R2b form part of the product-forming
channel. Analogously, the secondary reactions of the pressure dependent reactions R3 and R4
lead to products via the reactions R3a and R3b, respective R6a and R6b. With this, the total
product quantum yield of the photolysis of CHDO is the sum of the individual product
quantum yields across all channels k, where the index k=2, 3, 6 stands for the non-radiative
reactions R2, R3, and R6.
The individual product quantum yield can be approximated by
$$\Phi_k^{tot} = \frac{1}{1+a \cdot exp\left(\frac{\varepsilon_k - \varepsilon_0}{b}\right) \cdot \frac{[M]}{[M_0]}}$$
(F5)

analog to the publication by Röth and Ehhalt (2015) on $CH_2O$.
In equation F5, $\varepsilon_2$ is the excitation energy of the photolysis reaction. The energies $\varepsilon_3$ and $\varepsilon_6$
are related to $\varepsilon_2$ by the approximated energy transfer in a collision, respective by the averaged
width of the band intervals, given by $\varepsilon_3 = \varepsilon_2$ -0.0124 eV (Troe,2007) and $\varepsilon_6 = \varepsilon_2$ -0.13 eV
(Miller and Lee,1978). The pivot wavelength $1/\varepsilon_0$ is 348.6 nm, as published in Nilsson et al.

155  (2014).


The total quantum yield of the products (molecules plus radicals) can be deduced from the
rate constants of Table 2 and the measurements of Nilsson et al. (2010, 2014), who



investigated the pressure dependence of the kinetic isotope effect KIE of the photolysis
frequencies of $CH_2O$ and CHDO.
$$KIE = \frac{j_{CH2O}}{j_{CHDO}} \qquad \text{with} \quad j = \int \Phi^{tot} \sigma F \; d\lambda \qquad \text{(F6)}$$

As the quantum yield of $CH_2O$ is known from the literature (see e.g. Röth and Ehhalt, 2015)
$\Phi_{CHDO}^{tot}$ remains the only unknown factor in formula F6. With the actinic flux density F of the
lamp used by Nilsson et al. (2014) and the absorption spectra $\sigma_x$ of $CH_2O$ and CHDO from
Gratien et al. (2007) the ratio KIE can be calculated with optimized values for *a* and *b* in F5.
Comparing the results of the simulation with the measured data by Nilsson et al. (2010, 2014)
the constants *a* and *b* can be determined via a least square fit. Figure 2 presents the result with
optimal values *a*=2.94 and *b*=6.5×10$^{-5}$ nm$^{-1}$ together with measurements. The mean of the
data at 1000 hPa is included in the fit to account for the large variation of the data.

The total product quantum yield, deduced from the reaction scheme R0 to R7 is
$$\Phi^{tot} = \frac{k_2}{\alpha} \cdot \Phi_2^{tot} + \frac{k_3[M]}{\alpha} \cdot \Phi_3^{tot} + \frac{k_4[M]}{\alpha} \cdot \frac{k_6}{\beta} \cdot \Phi_6^{tot} \qquad \text{(F7)}$$

with α and β as defined in formula F3, and $\Phi_k^{tot}$, the sub-product yield, according to formula
F5. The measured wavelength dependence of $\Phi^{tot}$ at 1000 hPa pressure is depicted in Figure 3,
where the full circles represent the total quantum yield calculated with the rate constants from
Table 2. The pressure dependence of the three terms of $\Phi^{tot}$ is illustrated in Figure 4.

To obtain a smooth wavelength dependence, these rate constants can be represented by an
approximation function
$$k = A \exp (B (\lambda\text{-}300\text{nm})) \qquad \text{(F8)}$$

A least square fit gives the values for the parameters A and B listed in Table 3. If the value of
B was less than 0.001 it was set to 0. The wavelength dependence of $\Phi^{tot}$ at 1000 hPa with
these functions is presented by the solid line in Figure 3.

**Table 3:** Parameters of the rate constants according to equation F8, B in nm$^{-1}$ and A in s$^{-1}$,
respective in cm$^3$ s$^{-1}$.

|   | $k_1$ | $k_2$ | $k_3$ | $k_4$ | $k_5$ | $k_6$ | $k_7$ |
|---|---|---|---|---|---|---|---|
| **A** | 2.90 10$^5$ | 6.10 10$^8$ | 7.70 10$^{-10}$ | 1.30 10$^{-10}$ | 3.00 10$^5$ | 1.50 10$^8$ | 1.2 10$^{-12}$ |
| **B** | 0 | 0.086 | 0.069 | 0.071 | 0 | 0.075 | 0 |




For CHDO the only quantitative indication for the quantum yield of the radical channel in the
literature are measurements of the kinetic isotope effect KIE (Feilberg et al., 2007, Rhee et al.,
2008, Röckmann et al., 2010, and Nilsson et al., 2014). To simulate these KIE-measurements,
three parameters for the individual radical quantum yield $\Phi_k^{rad}$ are needed: the maximum
value $\Phi^{max}$ of the wavelength dependence, its curvature $b$, and the pivot wavelength $\lambda_0$ (here,
the parameter $a$ is 1). For the individual quantum yield no pressure dependence is assumed.
$$\Phi_k^{rad} = \frac{\Phi^{max}}{1+a\,exp\left(\frac{\varepsilon_k-\varepsilon_0}{b}\right)} \tag{F9}$$

Analog to the analysis for $CH_2O$ (Röth and Ehhalt, 2015), where the curvatures of the
wavelength dependence of $\Phi^{tot}$ and $\Phi^{rad}$ are similar, $b$ can be set to $6.5\times10^{-5}$ nm$^{-1}$ for the
radical quantum yield of CHDO. The maximum $\Phi^{max}$ was varied in the interval [0.70 / 0.78]
around the corresponding value for $CH_2O$, but the resulting scattering is very small (see
shaded area in Fig. 5). Consequently, parameter $\Phi^{max}$ is set to 0.74, matching the value also
used for $CH_2O$ (Ehhalt and Röth, 2015).

With these parameters the KIE of 1.63 as measured by Röckmann et al. (2010) was fitted with
the actinic flux density given by Röckmann et al. and the optical spectra by Gratien et al.
(2007). The best fit gave a pivot wavelength $\lambda_0$ of 327 nm. This value lies in the middle of
the bond energies of 362.63 kJ/mol for C-H and 369.6 kJ/mol for C-D, calculated by Chuang
et al. (1987). With the constants $\Phi^{max} = 0.74$, $a=1$, $b = 6.5\ 10^{-5}$ nm$^{-1}$ and $1/\varepsilon_0 = 327.1$ nm the
quantum yield function $\Phi^{rad}$ of the radical channel of CHDO is analog to F7:

$$\Phi^{rad} = \frac{k_2}{\alpha}\cdot\Phi_2^{rad} + \frac{k_3[M]}{\alpha}\cdot\Phi_3^{rad} + \frac{k_4[M]}{\alpha}\frac{k_6}{\beta}\cdot\Phi_6^{rad} \tag{F10}$$

where the radical quantum yields of the individual channels is given by function F9 and with
$\alpha$ and $\beta$ as defined in F3. Figure 5 depicts the wavelength dependence of the total quantum
yield together with that for the radicals.

To provide a more handy tool for atmospheric modeling, we introduce an exponential
function (F11), with only three parameters for the total and the radical quantum yields of
CHDO, similar to those deduced by Ehhalt and Röth (2015) for $CH_2O$, as a proxy for the 3-
term function F10:
$$\Phi = \frac{a}{1+exp\left(\frac{-\left(\frac{1}{\lambda}-\frac{1}{\lambda_0}\right)}{b}\right)\frac{[M]}{[M_0]}} \tag{F11}$$





The corresponding parameters for the total quantum yield of CHDO are a=1.0, b=7.7×10$^{-5}$ s$^{-1}$,
and λ$_0$=336.2 nm. For the radical channel the factor [M]/[M$_0$] is set to 1, as the photolysis
leading to the radicals is nearly pressure independent. The respective parameters are a=0.74,
b=7.7×10$^{-5}$ s$^{-1}$, and λ$_0$=325.0 nm. Both approximation curves are depicted in Figure 4, and
Figure 6 shows the pressure dependent comparison with the measured data by Miller and Lee

224    (1978).



**4. Analysis of the CHDO photo-decomposition**
Our second approach to estimate the quantum yields for the photolysis of CHDO is based on
the experiments of McQuigg and Calvert (1969) who measured the photo-decomposition of
$CH_2O$, CHDO, and $CD_2O$. Unfortunately, the authors only presented the quantum yields for
the two radical reaction channels of $CH_2O$ and $CD_2O$. They further assumed that the total
quantum yield equals 1, independent of wavelength. It appears, however, that these data have
a bias which becomes evident when the data for $CH_2O$ are compared to more recent
measurements.

In Figure 7 the dependence on the wavelength of $\Phi^{rad}$ of $CH_2O$ by McQuigg and Calvert
(1969) is depicted together with a curve for $CH_2O$, averaged over measured data from the
paper by Röth and Ehhalt (2015). The latter evaluation showed no pressure dependence, but
indicated a weak temperature effect which is neglected here. The curve is represented by the
following function:

$$\Phi^{rad}_{CH2O} = \frac{0.74}{1+exp\left(\frac{-\left(\frac{1}{\lambda}-\frac{1}{327.4}\right)}{5.4\times10^{-5}}\right)} - \frac{0.40}{1+exp\left(\frac{\frac{1}{\lambda}-\frac{1}{279.0}}{5.2\times10^{-5}}\right)}$$    (F12)

Equation F12 exhibits a maximum in $\Phi^{rad}$ around 310 nm, independent of the small
temperature shift, whereas the earlier values of McQuigg and Calvert exhibit a monotonic
decay with increasing wavelength above 280 nm, which points to a bias in the latter. The
second summand in F12 is less than 1% at wavelengths above 300 nm and, hence, can be
omitted in the present paper. Figure 7 also includes the data of McQuigg and Calvert (1969)
for $CD_2O$ which show a quite similar wavelength dependency as the data for $CH_2O$.

Our first assumption is that the bias in the experiments of McQuigg and Calvert extends
equally to both isotopologues ($CD_2O$ and $CH_2O$), and that, therefore, the ratio R of their





quantum yields is correct. This ratio is displayed in Figure 8 and shows a mostly monotonic
decrease with increasing wavelength. In this context, it is interesting to note that the ratio of
the rate constants for the decomposition of excited $CH_2O^*$ and $CD_2O^*$ into the respective
radical channels, as calculated by Troe (1984) from theory, result in a curve with a monotonic
decrease with increasing wavelength similar to that of the quantum yield ratio.
Using the ratio from Figure 8 together with the fit function F12 for $\Phi_{CH2O}^{rad}$ allows to estimate
$\Phi_{CD2O}^{rad}$ for the radical channel of $CD_2O$, as shown in Figure 8.
To calculate $\Phi_{CHDO}^{rad}$ we need one further assumption. Our hypothesis is suggested by the
results of Feilberg et al. (2004), who found that the KIE of the reactions of CHDO with OH,
Cl and Br are arithmetic means of the KIE of the reactions of $CH_2O$ and $CD_2O$ with those
radicals. This in turn means that the bond strengths for C–H, respectively C–D remain nearly
the same in the different isotopologues. We, therefore, assume that $\Phi_{CHDO}^{rad}$ can be calculated
from the average of $\emptyset_{CH2O}^{rad}$ and $\emptyset_{CD2O}^{rad}$ at each wavelength:
$$\Phi_{CHDO}^{rad}(\lambda) = \left( \Phi_{CH2O}^{rad}(\lambda) + \Phi_{CD2O}^{rad}(\lambda) \right)/2 \qquad \text{(F13)}$$
The quantum yields are compared in Figure 9. $\Phi_{CHDO}^{rad}$ does not depend on pressure since
$\Phi_{CH2O}^{rad}$ nor $\Phi_{CD2O}^{rad}$ are pressure dependent. The respective maxima in $\Phi^{rad}$, on the other hand,
decrease from 0.72 over 0.70 to 0.65 for increasing deuteration. Moreover, there is a blue shift
of 5 nm, resp. 10 nm in the decreasing part of the quantum yield spectra of CHDO and $CD_2O$,
i.e. at wavelengths above 315 nm. These blue shifts have the same tendency but do not quite
match the measured threshold energies of 362.3 kJ/mol, 368.4 kJ/mol, and 370.6 kJ/mol for
$CH_2O$, CHDO, and $CD_2O$, respectively (Chuang et al.,1987), which correspond to the
wavelengths 330.9 nm, 325.5 nm, and 323.5 nm.

The 1-Term fit function for the radical channel of CHDO is:
$$\Phi_{CHDO}^{rad} = \frac{0.72}{1 + exp\left( \frac{-\left( \frac{1}{\lambda} - \frac{1}{323.0} \right)}{7.7 \times 10^{-5}} \right)} \qquad \text{(F14)}$$
In Figure 10 the result of the interpretation of the measured photo-decomposition of CHDO
by McQuigg and Calvert (1969) is compared to the radical quantum yield deduced from the
fluorescence measurements of Miller and Lee (1978). Both estimations lead to a wavelength
dependence of $\emptyset_{CHDO}^{rad}$ which lie in each others uncertainty range. This is a strong hint that the
deduced results are robust and represent the true quantum yield of the radical channel of the
photolysis of CHDO.






**5. The isotope fractionation during the photolysis of CH₂O**


The photolysis frequency $J_i$ of the isotopologues $CH_2O$ and $CHDO$ is given by the integration
of quantum yield $\Phi$, absorption cross section $\sigma$, and spectral actinic photon flux density $F_\lambda(\lambda)$
over the $\lambda$ wavelength domain:
$$J_i = \int \varphi_{i,j}(\lambda) \cdot \sigma_i(\lambda) \cdot F_\lambda(\lambda) \, d\lambda \tag{F15}$$

where the quantum yield $\Phi_{i,j}(\lambda)$ depends on the product channel j, either molecular or radical,
of the isotopologue i, and the absorption cross section $\sigma_i(\lambda)$ is specific to the isotopologues i.
For our calculations the absorption spectra of $CH_2O$ and $CHDO$ from Gratien et al. (2007)
were applied. We used these values instead of the JPL-recommendation (Burkholder, 2020)
for consistency with the calculations in section 2 and 3. The solar spectral actinic flux density
$F_\lambda$ was calculated from a quasi-spherical 1-D radiation transfer model (Röth, 2002); the $\Phi(\lambda)$
are those from section 2. An example of the terms $\Phi(\lambda)$, $\sigma(\lambda)$, $F_\lambda(\lambda)$ for the molecular channel
of CHDO is given in Figure 11 for the pressure and temperature at an altitude of 20 km. The
product of these terms, integrated over 5 nm intervals for better visibility, is also displayed to
demonstrate the spectrally resolved contributions to the photolysis frequency of the molecular
channel of CHDO.

The kinetic isotope effect for the molecular channel is given by
$$KIE_{mol} = \frac{J_{CH2O}^{mol}}{J_{CHDO}^{mol}} \tag{F16}$$

and correspondingly for the radical channel
$$KIE_{rad} = \frac{J_{CH2O}^{rad}}{J_{CHDO}^{rad}} \tag{F17}$$

For a quick overview the dependence of $KIE_{rad}$ and $KIE_{mol}$ on altitude for globally averaged
conditions (equinox, 30°N) are depicted in Figures 12a and 12b. $KIE_{mol}$ decreases
monotonically with decreasing pressure from 1.59 at 1000 hPa to 1.06 at 1 hPa. The radical
channel in contrast shows hardly any pressure dependency as the rate of this reaction is not
influenced by the quenching process. The marginal variation of the kinetic isotope effect with
altitude is caused by the altitudinal increase of the photon flux and its differing contribution to
the photolysis frequency integrals of $CH_2O$ and $CHDO$.

To examine whether the quantum yield functions for CHDO deduced above are applicable for
modeling purposes, additional sensitivity studies were carried out, varying the main features





of the quantum yield functions. With respect to the fractionation factor, only the variations of
those parameters are relevant which alter the relation of the entire photolysis frequency
integrals (eq. F15) of the molecular and the radical channels. In Figures 12a and 12b we
additionally show the variances of the photolysis frequencies as well as of the fractionation
factors. The shaded area is produced by varying one parameter of the CHDO quantum yield
within a range of roughly ±10%, as indicated below. The photolysis frequency of $CH_2O$
remained unchanged.

The photolysis frequency of the radical channel of CHDO is only sensitive to the maximum of
the quantum yield and to the threshold wavelength 323 nm. Shifting the latter value by ± 3 nm
produces changes of about 20 % in the troposphere, decreasing to 10 % at 50 km altitude as
shown in Figure 12a. This variation of the threshold produces an error bar of the fractionation
factor of the same magnitude.

The sensitivity of the molecular branch of the photolysis frequency of CHDO to the
preexponential factor of the quantum yield function is roughly 10 % throughout the
atmosphere if this value is varied by 10%. All other parameters do not alter the integral
equation F15 significantly and produce only variances less than 1 %. It can thus be concluded
that the estimated equation parameters are good representations of the actual values.

At higher altitudes (<10 hPa) $\Phi_{mol}^{CHDO}$ and $\Phi_{mol}^{CH2O}$ are close to unity in the wavelength regime
330 nm to 360 nm (see e.g. Fig. 6). So, the photolysis frequency in the stratosphere does not
change much if the parameters of the respective functions are varied. Therefore, the variance
of the fractionation factor does not much decrease above 30 km altitude. Here, measurements
at tropospheric pressures could be much more informative as becomes evident from Figure
12b.


**6. Discussion**

Up to now there had been a handicap in the interpretation of stratospheric measurements of
the concentration of deuterated hydrogen HD due to the lack of exact knowledge of the
photolysis frequencies of deuterated formaldehyde, resulting in an uncertainty on the
fractionation factor. There have been a number of experimental approaches to deduce the



fractionation factor, where e.g. Feilberg et al. (2005) measured a value of 1.82 ±0.07 for $\alpha_{mol}$
Röckmann et al. (2010) found a value of 1.63 ± 0.03 for that ratio. In a modeling paper, Mar
et al. (2007) varied the fractionation factor between 1.2 and 1.5 for stratospheric conditions.

In all these studies the pressure dependence of the photolysis frequencies could not be
investigated. An interesting experiment by Nilsson et al. (2009) addressed this problem.
Unfortunately, the spectral radiance of the light source used did not resemble the sun light
well enough, and their findings could not be transferred to the real atmosphere without
information on the quantum yield of CHDO.

Beside its pressure dependence the variation of the photolytic fractionation factors can also be
caused by different actinic fluxes at the times and sites of the experiments. The actinic flux in
the numerator and denominator of the fractionation factor in equations F16 and F17 do not
cancel out, and, therefore, the factor is depending on the local insolation conditions.
Calculations of the solar zenith angle (SZA) dependency with the complex radiation transfer
model ART (Röth, 2002) result in values from 1.47 at overhead sun to 1.95 at SZA=83° for
clear sky and free horizon at ground level. This zenith angle dependency is less expressed at
20 km altitude and disappears at 50 km, as depicted in Figure 13. This effect may explain the
differences in the measurements of the fractionation factors. To check the variance with the
solar zenith angle the measured fractionation factor  $KIE_m$ (eqs. F16 and F17) is compared to
model calculations. The factor 1.63 ±0.03 (Röckmann et al., 2010) was derived from
experimental studies in the atmospheric simulation chamber SAPHIR between 60° and 70°
SZA (Röckmann et al., 2010). The absorption cross sections by Gratien et al.(2007) and the
quantum yields derived above together with the radiation spectra result in a fractionation
factor of 1.54 for 60° SZA and 1.70 for 70° SZA are in good agreement with the measured
value.

**Conclusions**

The current work derives a framework and set of equations for describing the CHDO
photolysis, based on two different approaches building on the available literature data, finding
a consistent result across all data sets. It could be shown that the most influential parameters
of the rates of photolysis of CHDO are the absolute value and the threshold of the quantum
yield of the radical channel. Simplified equations (F11 and F14) that are readily implemented



in kinetic models are provided for these quantities. Measurements around 300 nm and 325 nm
could further reduce the uncertainty on the fractionation factor. Additional measurements of
the pressure dependence of the total quantum yield, i.e. the quenching rate of excited $CHDO^*$,
would be valuable to further test the assumptions made in this paper.

**Competing interests**
The contact author has declared that none of the authors has any competing interests


**Acknowledgments**
The authors thank Dr. B. Bohn and Dr. D. Tarraborrelli for their useful comments and
suggestions to improve the clarity and readability of the paper.



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



## Figures

**Fig01** : Comparison of the measured data by Miller and Lee (1978) (full dots) with the
pressure dependent part of the fitted function fkt(M]) for different wavelengths in nm as
indicated.

**Fig02 :** Pressure dependence of the photolysis frequency ratio α of $CH_2O$ and CHDO
compared to the measured data of Nilsson et al., 2010 (blue squares) and Feilberg et al., 2007,
Rhee et al., 2008, Röckmann et al., 2010 (red squares, 'others')

**Fig03:** Comparison of total product quantum yields calculated with the measured rate
constants of Miller and Lee (1978) (black circles) with the fit function F8 and the reaction rate
parameters of Table 3 (solid curve).

**Fig04:** Wavelength dependence of the contributions of the 3 terms of equation F7 to the total
quantum yield of the CHDO photolysis at 10 hPa (a) and 1030 hPa (b).

**Fig05:** The total quantum yields of the photolysis of CHDO  and that of the radical channel
calculated with the 3-Term function F8 (blue curves). The shaded area indicates the variation
of parameter a within the interval [0.70 / 0.78]. The red curves represent the 1-Term
approximation functions.

**Fig06:** Comparison of the 1-Term fit function (open circles with solid line) with the measured
data (Miller and Lee, 1978) of the total photolytic quantum yields (full circles) at 0, 10, 200,
and 1000 hPa.

 **Fig07:**  The original data of McQuigg and Calvert (1969) for $CH_2O$ (full red squares) and
$CD_2O$ (open squares) in comparison with the averaged function for $CH_2O$ by Röth and Ehhalt
(2015) for the photolytic quantum yield of the radical channels.

**Fig08:** The ratio $\Phi_{CD2O}/\Phi_{CH2O}$ of the data from Fig.7 and the corrected radical quantum yield
of $CD_2O$. The triangles depict the theoretical data of Troe (1984) for the ratio of the respective
reaction constants. The black squares are the corrected quantum yields for $CD_2O$ (see text).





**Fig09:** The wavelength dependency of the quantum yields for the radical channel of the
isotopologues of formaldehyde. The curve for $CH_2O$ is from Röth and Ehhalt (2015), that for
$CD_2O$ represents the corrected data of McQuigg and Calvert (1969), and the black dots for
CHDO are the mean of both. Included is also the fit function for CHDO.

**Fig10:** The CHDO quantum yield fit functions of the deduction from the fluorescence
measurements (blue) of Miller and Lee (1978) and for the interpretation of the photo-
decomposition (red) measurements of McQuigg and Calvert (1969). Also depicted is the
function of the total quantum yield.

**Fig11 :** Contributions to the molecular channel of the photolysis of CHDO at 20 km altitude
to the photolysis rate integrated over 5 nm wavelength, by the actinic photon flux, the
respective quantum yield, and the absorption cross section (Gratien et al., 2007). The
photolysis rate, the cross section, and the photon flux are multiplied by $2.5 \times 10^5$ sec, $1.5 \times 10^{19}$
$cm^{-1}$, and $2.5 \times 10^{-15}$ photons$^{-1}$ nm sec, respectively.

**Fig12a :** Altitudinal dependency of the photolysis frequencies of the radical channel of $CH_2O$
and CHDO. Also shown is the ratio of these values.

**Fig12b :** Altitudinal dependency of the photolysis frequencies of the molecular channel of
$CH_2O$ and CHDO. Also shown is the ratio of these values.

**Fig13 :** The solar zenith angle dependency of the photolysis frequency ratio of the molecular
channel at different altitudes.







Fig. 1






Fig. 2




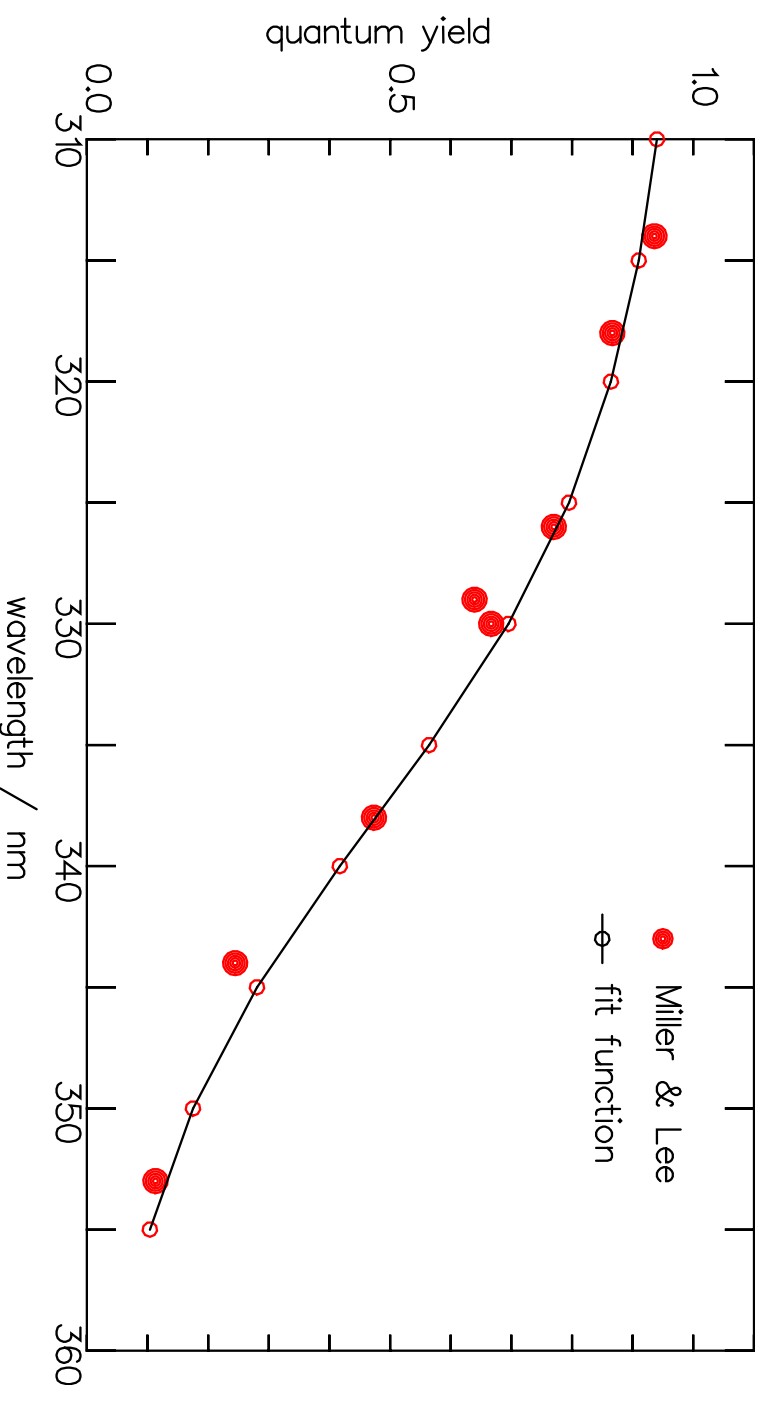


Fig. 3






640                    Fig. 4






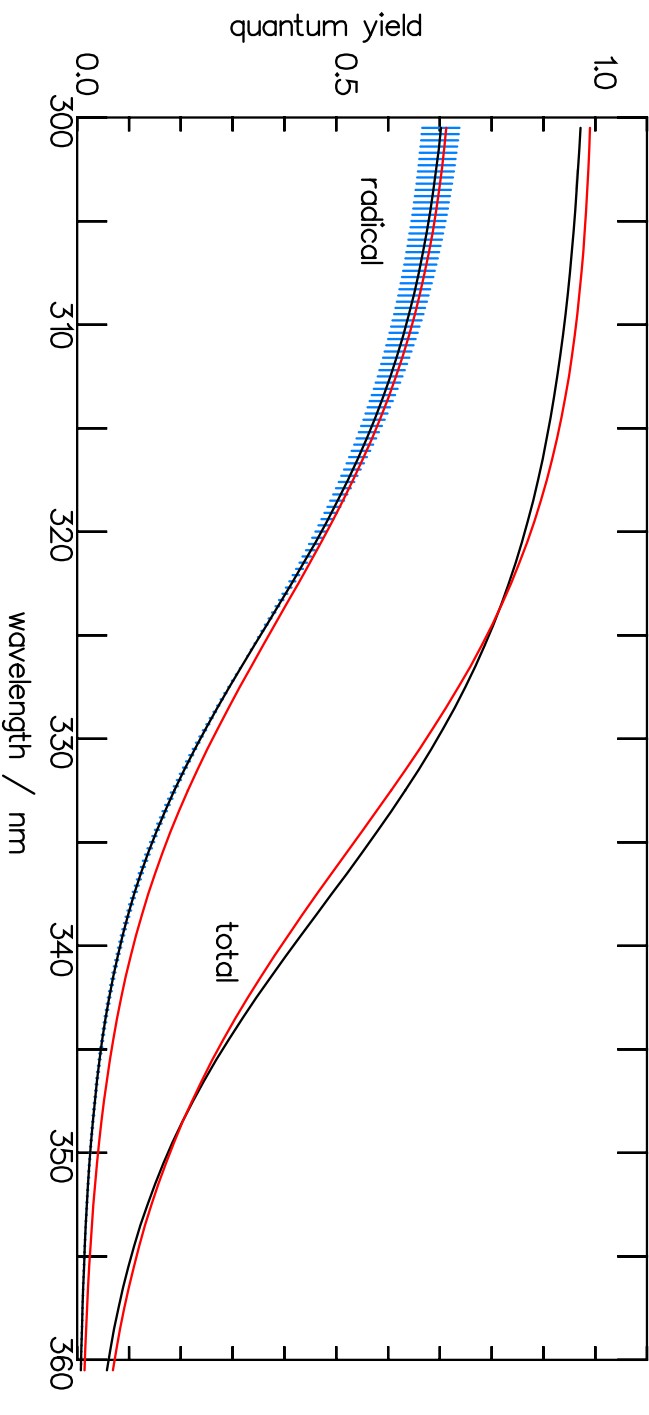


Fig. 5



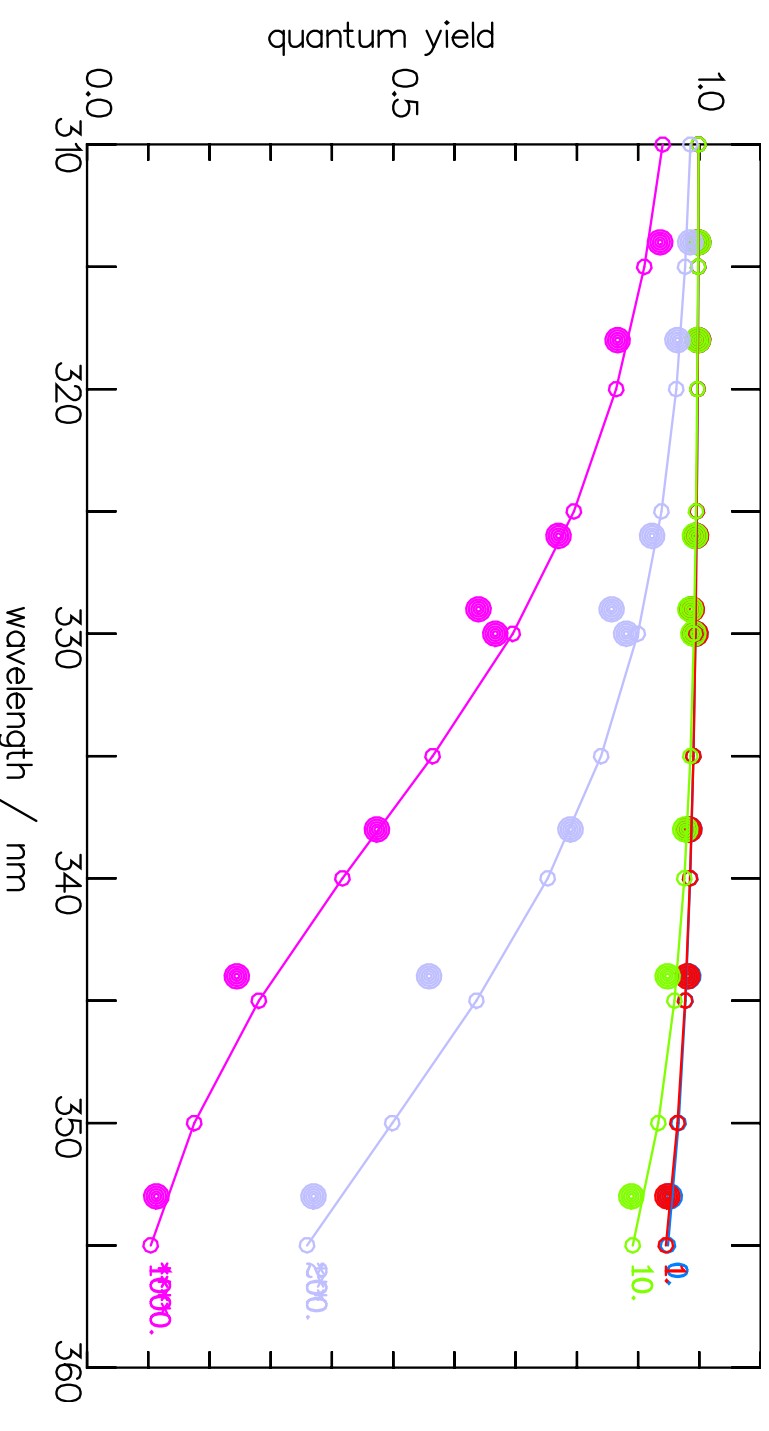

Fig. 6





Fig 7



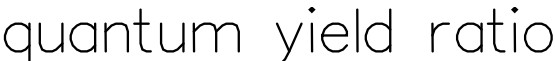


Fig. 8





Fig. 9




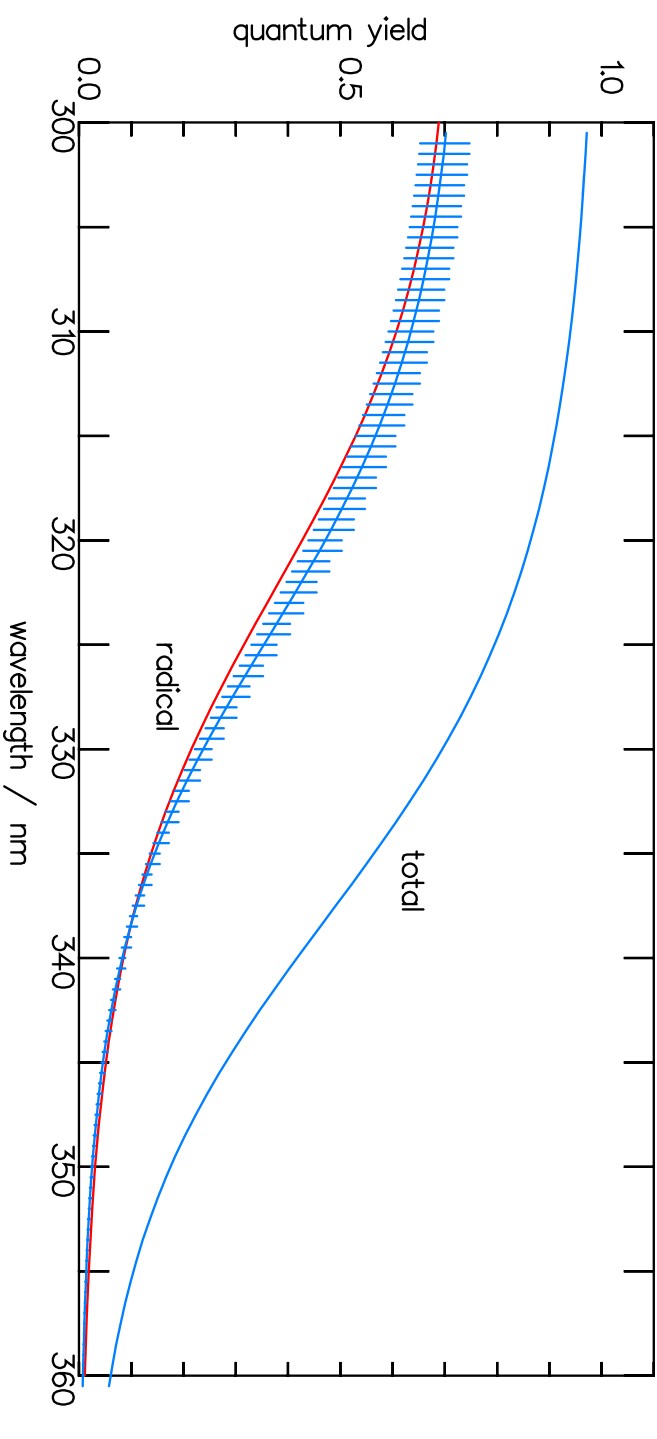

664        Fig. 10




Fig. 11







Fig. 12a

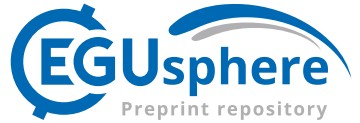






Fig. 12b









Fig. 13





$$CHDO(S_0) + h\upsilon \rightarrow CHDO^*(S_1) \qquad\qquad (R0)$$
$$CHDO^*(S_1) \rightarrow CHDO^\# + h\nu_1 \qquad\qquad (R1)$$
$$CHDO^*(S_1) \rightarrow CHDO^*(S_0) \qquad\qquad (R2)$$
$$CHDO^*(S_0) \rightarrow H + CDO/D + CHO \qquad\qquad (R2a)$$
$$CHDO^*(S_0) \rightarrow CO + HD \qquad\qquad (R2b)$$
$$CHDO^*(S_0) + M \rightarrow CHDO^\#(S_0) \qquad\qquad (R2c)$$
$$CHDO^*(S_1) + M \rightarrow CHDO^{*-\Delta\varepsilon 1}(S_0) \qquad\qquad (R3)$$
$$CHDO^{*-\Delta\varepsilon 1}(S_0) \rightarrow H + CDO/D + CHO \qquad\qquad (R3a)$$
$$CHDO^{*-\Delta\varepsilon 1}(S_0) \rightarrow CO + HD \qquad\qquad (R3b)$$
$$CHDO^{*-\Delta\varepsilon 1}(S_0)\, M \rightarrow CHDO^\#(S_0) \qquad\qquad (R3c)$$
$$CHDO^*(S_1) + M \rightarrow CHDO^{*-\Delta\varepsilon 2}(S_1) \qquad\qquad (R4)$$
$$CHDO^{*-\Delta\varepsilon 2}(S_1) \rightarrow CHDO^\#(S_0) + h\nu_5 \qquad\qquad (R5)$$
$$CHDO^{*-\Delta\varepsilon 2}(S_1) \rightarrow CHDO^{*-\Delta\varepsilon 2}(S_0) \qquad\qquad (R6)$$
$$CHDO^{*-\Delta\varepsilon 2}(S_0) \rightarrow H + CDO/D + CHO \qquad\qquad (R6a)$$
$$CHDO^{*-\Delta\varepsilon 2}(S_0) \rightarrow CO + HD \qquad\qquad R(6b)$$
$$CHDO^{*-\Delta\varepsilon 2}(S_0) + M \rightarrow CHDO^\#(S_0) \qquad\qquad (R6c)$$
$$CHDO^{*-\Delta\varepsilon 2}(S_1) + M \rightarrow \; \ldots \qquad\qquad (R7)$$


Reaction scheme (Table 1)