# Peer review of "Quantum Yields of CHDO above 300 nm"

_EGUsphere, 2023_

## Author Comment (AC1)

We thank the referees for their thoughtful comments, which helped improve the paper. Below, we summarize the modifications made to the document based on the specific comments of the referees. Both referees had suggestions regarding the presentation of the material (self-sufficient captions, consistency of notation,..). In response, we have made many edits to the manuscript to ensure the various sections, captions and tables cross-reference well and use consistent notation. These typographical edits are not indicated in detail below.

** Answer to the comments of Referee #1

This paper brings life to old data from which consistent formulations of the molecular and radical channel quantum yields in CHDO photolysis are extracted. The results probably merit publication after considering a number of comments.

In general, the Figure captions appear to be of the "insider type" and not such that the Figures can be viewed and understood by the reader without a meticulous inspection of the text. The criticism applies to most of the figures; taking Figure 1 as example: "Comparison of the measured data byMillerand Lee (1978) (full dots) with the pressure dependent part of the fitted function fkt(M]) for different wavelengths in nm as indicated." Without consulting the text, the reader is left wondering: Which data? Which function? What is Q(M)?

*We have updated all figure captions, making them self-sufficient in as much as possible, and re-using the notation of the main text to facilitate cross-referencing with the discussion.*

Table 1 is not "self-consistent" in the sense that all symbols are defined either in the Table caption or in footnotes (De1 and De2).

*Table 1: '-$\Delta\varepsilon_1$, -$\Delta\varepsilon_2$ indicate the energy losses in the respective reactions' is added to the table caption. We also note that the scheme is a two-step representation of a multi-step quenching process.*

The first Figure referenced in the text is Figure 4a in Section 3 (line 106). It is as if the authors have done a last-minute paragraph swap. Caption to Figure 4a: "Wavelength dependence of the contributions of the 3 terms of equation F7 to the total quantum yield of the CHDO photolysis at 10 hPa (a) and 1030 hPa (b)." At this point in the text, the reader is about to be introduced to where the rate coeficients k , k , k and k origin (that is, Table X in the paper byMillerand Lee, 1978). The rate coeficients and the derived rate coeficients (k , k , k and k ) are then summarized in Table 2. The authors add confusion by citing Table 2 ofMillerand Lee (it is actually TABLE II) and then their own Table 2 a few lines later.

*At that point it is indeed a bit early to refer to figure F4a. The reference was removed and replaced by '(see later)'. The text now uses "Table X" and "Table II" with explicit reference to the original data.*

Equation (F4) is a pseudo-equation that does not add to the value of the paper and it would perhaps be better write equation (F3) in the form below and maybe indicate the wavelength dependency as well:   [ Equation ]

*Equation F4 is completed. It is needed explicitly as it is plotted in Figure 1.*

Table 2. The excitation wavelengths listed are rounded numbers from theMillerand Lee (ML) 1978 publication that states: the out-of-plane bending mode (n ) is the inducing mode for radiative transitions and also the promoting mode for the nonradiative transitions. Why not use the wavelengths given by ML ? Why are the 316.2 nm ML data not used? The ML 320.2 nm data are entered in Table 2 with 329 nm excitation – why? Have the authors located misprints in the ML TABLE II? If yes, this should be communicated. The 329, 344 and 353 nm data for k and k in Table 2 do not correspond to any t-numbers in the ML TABLE II. Where do these numbers come from? The 329 nm data (ML 320.2 nm) refers to fluorescence from the 1 4 (S ) state (n corresponds to the CH str), and the "next lower vibrational level" would be the 4 (S ) state, for which to fluorescence data are found for 353 nm excitation. However, this is the only case not involving the n mode (C=O str). Obviously, there are no data for the "next lower vibrational levels" of the 4 and 4 states involving n , yet numbers appear for k and k in Table 2. In summary, the selection of data could be better described.

*The wavelength 329 nm seems to be an outlier, and even the authorsMillerand Lee did not use these data later in their paper. We removed that entry from the table, now stating explicitly that we use the $2^i 4^j$ transitions only. The table is upgraded to show the exact wavelengths from the original publication. For 344.4 and 352.9 nm, $k_5$ and $k_6$ are now indicated explicitly as estimates.*

Line 154. The authors state: The pivot wavelength $1/\varepsilon$ is 348.6 nm, as published in Nilsson et al. 155 (2014). Maybe they should mention that the 348.8 nm origins from quantum chemistry calculations of the barriers to dissociation H-CHO, H-CDO, D-CHO and D-CDO.

*We now mention explicitly that these come from theoretical calculations, adding the text '…from quantum chemistry calculations of the barriers to dissociation H-CHO, H-CDO, D-CHO, and D-CDO'*

Line 168. "optimal values" should probably be "optimized values"

*'optimal values' is replaced by 'optimized values'.*

Table 3, Line 185. The A-value for k must have been derived taking the k -value for 329 nm as an outlier. Are there other examples of data-massage? There are no error estimates included in Table 3 to indicate the validity of the smoothing procedure.

*The 329 nm value is an outlier and indeed omitted. See also our comments on changes to Table 2 above, where we now state explicitly we only use the $2^i 4^j$ transitions, and have 2 rows containing estimates for k5 and k6. No other data-massage was performed, other than neglecting the B-factor if the wavenumber-dependence is negligible (see text below eq. F8).*

*The caption for Table 3 now also states the values are derived from a least square fit, and the related text below eq. F8 was updated for clarity. Finally, the estimated variance is added.*

Line 242. The numbers in the equation are not the same as those given in the reference, but are apparently rounded. However, some of the numbers are rounded outside the error limits given in the reference.

*Equation F12 is exactly the same as the recommended function (8) in Table 1 (p. 7199) of Röth and Ehhalt (2015) (omitting the uncertainties), which however does differ from the fitting equation (3) given earlier in the paper on p. 7196. The difference is due to the fitting procedure for radical quantum yields separately, versus radical+molecular+total quantum yields simultaneously.*

Figure captions, Line 562. As mentioned, the Figures including captions should preferably be self-consistent. In general, this is not the case.

*All figure captions are extensively rephrased, re-using the symbols as in the main text and referring explicitly to the equations underlying the graphs. We feel the captions are now significantly more informative.*

---

## Author Comment (AC2)

** Answer to the comments of Referee #2

The photolysis of formaldehyde is a major source of hydrogen throughout the atmosphere. Since the photolysis rates of formaldehyde and its deuterated analogues, pimarily CHDO, in their molecular channels are altitude dependent, there is an isotopic enrichment of HD relative to H2 in the stratosphere. Quantification of this enrichment is a long standing issue in atmospheric science. Based on previous experimental data on the ffuorescence as well on the photo-decomposition of formaldehyde and its deuterated analogues, together with detailed molecular kinetic modelling, the present paper provides an improved understanding of the wavelength dependence of molecular and radical channels of the photolysis of formaldehyde which goes beyond previous interpretations. The paper therefore is valuable and hence publishable provided that a number of aspects are taken into account:

In the summary stronger emphasis of the present findings with respect to their atmospheric relevance should be made in order to attract stronger interest of the average readership of this journal.

*The emphasis on the atmospheric relevance of the photolysis of formaldehyde is expressed by adding the sentence 'The importance of the photolysis of formaldehyde in the atmosphere is exhibited by presenting the altitudinal dependence of the isotopic fractionation through the yield of the HD channel' to the abstract.*

The organisation of the manuscript has a number of deficits which should be rectified. The first one is the presentation of the overall mechanism to include both radiative as well as collisonal deactivation steps. For instance: There is energy transfer (quenching) allowed in both S0 and S1 electronic states. But why is no fractionation of CHDO* – either via a molecular or radical channels - from the electronically excited S1 state included? Is the initial energy insuffcient?

*Most of the photolysis energy was used to reach the S1 state. The dissociation of the $S_1$ state itself needs additional energy (see e.g. the energy diagram in Chuang et al, Fig 1)*

The fact that the radical channels are pressure independent, as stated later in the manuscript, should be indicated in the mechanism. In the form of the mechanism presented here, reactions (2a), (3a) and (6a) imply that this is not the case.

*That there is no pressure dependence at atmospheric conditions is now stated explicitly: 'At atmospheric pressures, as considered in this paper, the contributions of the individual quenching processes are insignificant with respect to the overall radical quantum yield'.*

The inclusion of a schematic energy diagram of both electronic states involved and their threshold energies for decomposition together with arrows for the different pathways of excitation und de-excitation via the cascade would significantly improve the understanding of the mechanism.

*As we are not working with explicit energy levels, but rather with grouping based on an above/below dissociation threshold criterium, we refrain from showing the energy diagram and refer instead to the literature.*

In such mechanisms with energy dependent channels the fundamental molecular rate coefficients are principally k(E)s as delineated in detail in the paper by Troe. These become thermal rate coefficients k only with the underlying assumption that the system is thermalized at all energies. Very likely this is the case under atmospheric conditions, but it should be mentioned.

*It is now mentioned explicitly that the system is thermalized under atmospheric conditions.*

In the form presented in the manuscript, all quenching reactions are not mass conserving because the collider M does not show up on the product side of the equations. Please correct.

*M is added to the product side where necessary. (Table 1)*

It should also be mentioned that due to this mechanism and the consecutive reactions of the products only the molecular channel contributes to isotopic fractionation. The products of the radical channel (H+CHO) never end up as H2.

*The sentence 'Due to consecutive reactions only the molecular channel contributes to the HD production.' is added to the text. The HD channel is also mentioned in the abstract.*

On line 262 it is stated that the bond strength of C-H and C-D are almost the same. In the view of the reviewer this is not consistent with the notion that the zero point energy for C-D is lower than that of C-H. It also contradicts the statement in this paper that the „threshold energies for the radical channels in CH2O and CHDO are different (lines 271-273)"

*Our original formulation can indeed be misunderstood. To avoid this confusion, the sentence is rephrased to 'This in turn implies that the C-H bond strengths are similar in the isotopologues, and the same is true for the C-D bond strengths.'*

The correspondence between text, figure captions and figures needs much better organisation in order to improve the readability of the manuscript. For instance, Fig. 4 should never be the first figure to be cited in the text. Moreover, the figure captions should be more self-explanatory and more comprehensible even without recurrence to the text

The main text, as well as the figure and table captions, have been modified to provide a more readable and self-contained text, with consistent use of symbols and cross-referencing to the equations where needed. Fig. 4 is now no longer cited first (at that point it the content of Fig 4 was not explained yet), in favor or a "see below" reference.

In line 266 is a „neither" missing

*'neither' is added.*

In line 275 replace „1-term" by „one-term"

*'1-term' and '3-term' are replaced by 'one-term' and 'three-term' everywhere. The words have also been inserted in text and figure captions to help differentiate between the various functions.*

The chapter headline „Isotopic fractionation during the photolysis of CH2O" should be extended to include CHDO

*In the chapter headline 'CH$_2$O' is replaced by the more general term 'formaldehyde'.*

Line 606-608 figure caption of Fig. 11 needs rephrasing.

*The figure caption of Fig.11 is rephrased, re-arranging the text and explicitly providing the necessary symbols to link to the main text.*